# Ginsenosides from *Panax ginseng* as Key Modulators of NF-κB Signaling Are Powerful Anti-Inflammatory and Anticancer Agents

**DOI:** 10.3390/ijms24076119

**Published:** 2023-03-24

**Authors:** Won Young Jang, Ji Yeon Hwang, Jae Youl Cho

**Affiliations:** Department of Integrative Biotechnology, Sungkyunkwan University, Suwon 16419, Republic of Korea

**Keywords:** NF-κB, inflammatory illnesses, cancer, *Panax ginseng*, ginsenosides

## Abstract

Nuclear factor kappa B (NF-κB) signaling pathways progress inflammation and immune cell differentiation in the host immune response; however, the uncontrollable stimulation of NF-κB signaling is responsible for several inflammatory illnesses regardless of whether the conditions are acute or chronic. Innate immune cells, such as macrophages, microglia, and Kupffer cells, secrete pro-inflammatory cytokines, such as TNF-α, IL-6, and IL-1β, via the activation of NF-κB subunits, which may lead to the damage of normal cells, including neurons, cardiomyocytes, hepatocytes, and alveolar cells. This results in the occurrence of neurodegenerative disorders, cardiac infarction, or liver injury, which may eventually lead to systemic inflammation or cancer. Recently, ginsenosides from *Panax ginseng*, a historical herbal plant used in East Asia, have been used as possible options for curing inflammatory diseases. All of the ginsenosides tested target different steps of the NF-κB signaling pathway, ameliorating the symptoms of severe illnesses. Moreover, ginsenosides inhibit the NF-κB-mediated activation of cancer metastasis and immune resistance, significantly attenuating the expression of MMPs, Snail, Slug, TWIST1, and PD-L1. This review introduces current studies on the therapeutic efficacy of ginsenosides in alleviating NF-κB responses and emphasizes the critical role of ginsenosides in severe inflammatory diseases as well as cancers.

## 1. Introduction

Inflammation is a pre-emptive defense mechanism against infections to protect the host’s body [1]; however, the dysregulation of inflammatory responses contributes to inflammatory diseases causing pain, fatigue, or even life-threatening symptoms, such as organ failure [2,3]. Inflammatory diseases include acute injury in various organs due to severe infection, which may lead to septic shock, and chronic type 2 inflammatory diseases, such as asthma, chronic obstructive pulmonary disease (COPD), and atopic dermatitis, often caused by long-term or allergic inflammation [4,5].

Nuclear factor kappa B (NF-κB) was first discovered by Ranjan Sen and David Baltimore in 1986 [6]. As its name suggests, NF-κB binds to the immunoglobulin κ light chain enhancer of activated B cells, regulating the induction of several target genes involved in the cell cycle, differentiation, activation, and apoptosis [7,8]. There are five different NF-κB subtypes: RelA (p65), RelB, c-Rel, p105 (which can be further cleaved and matured into p50), and p100 [9]. In particular, p65 and p50 are involved in the canonical activation of the NF-κB signaling pathway, regulating innate immunity by forming a dimer with each other [10]; however, the excessive and uncontrolled activation of NF-κB triggers the progression of inflammatory diseases as well as the occurrence of serious inflammatory lesions [11].

The canonical NF-κB signaling pathway, which activates NF-κB subunits p50 and p65, begins as pattern recognition receptors (PRRs) from macrophages interact with pathogen-associated molecular patterns (PAMPs) or damage-associated molecular patterns (DAMPs) [12,13]. Among various PRRs, toll-like receptors (TLRs) promote two different adapter molecules: myeloid differentiation primary response 88 (MyD88) and TIR-domain-containing adapter-inducing interferon-beta (TRIF) [14,15]. MyD88 and TRIF upregulate the kinase activity of the inhibitor of NF-κB kinase (IKK), which in turn degrades the inhibitor of NF-κB kinase (IκBα) and releases p50 as well as p65 [16]. Released NF-κB subunits translocate to the nucleus and stimulate M1 macrophage polarization by upregulating the gene expression of agents, such as cyclooxygenase-2 (COX-2) as well as inducible NO synthase (*i*NOS), and pro-inflammatory cytokines, such as interleukin (IL)-1, IL-6, and tumor necrosis factor-alpha (TNF-α) [17,18,19]. NF-κB also promotes the activation of the NLR family pyrin domain containing 3 (NLRP3) inflammasome by mediating the priming signal of the inflammasome [20]. The NLRP3 inflammasome is responsible for the maturation and secretion of pro-inflammatory cytokines, eventually leading to CD4+ T cell differentiation and activation [21].

Since the NF-κB pathway is deeply connected to the pathogenesis of various inflammatory diseases, many immunosuppressive or anti-inflammatory agents typically target the NF-κB pathway [22]. For instance, glucocorticoids such as dexamethasone induce IκBα expression in monocytes and block the binding of NF-κB to target DNA sites [23]. Nonsteroidal anti-inflammatory drugs (NSAIDs) suppress the function of COXs induced by the transcriptional activity of NF-κB [24]; however, glucocorticoids often exert serious side effects, such as muscle atrophy, whereas NSAIDs show a narrow application in that they only inhibit the downstream signals of NF-κB [25]. Moreover, it has been reported that NSAIDs upregulate TNF-α production, such that they can be used as a painkiller but not as a fundamental treatment [26]. Therefore, many studies have focused on natural compounds, which may show diverse efficacies on inflammatory diseases with small side effects. For instance, the anti-inflammatory effects of natural phenolic compounds, such as flavonoids, have been well studied.

Ginseng, a herbal plant in the genus *Panax*, mostly found across Asia, has been used as a traditional medicine for thousands of years. Among them, *Panax ginseng*, also known as Korean ginseng, has significant pharmacological activities against cancer, depression, aging, fatigue, and diabetes [27,28,29,30,31,32]. *Panax ginseng* also has mild side effects, but has little toxicity in normal doses [33]. It consists of various chemicals, including polysaccharides, peptides, vitamins, oils, flavonoids, and ginsenosides [34,35,36,37,38,39,40,41].

Triterpenes, precursors to steroids, are significantly included in plants and animals. Ginsenosides, triterpene saponins, contain 30 C atoms composed of six isoprene units and are characterized by backbone structures (Figure 1), (Table 1) [42]. Most of them have a dammarane skeleton, and they are classified into protopanaxadiol (PPD) and protopanaxatriol (PPT) types based on the position and number of hydroxyl groups. Compared to PPD, the PPT structure has a sugar side chain at carbon-6 (C6) and remains free at carbon-3 (C3) [43]. PPD has sugar attached to either C3 or carbon-20 (C20), while PPT has saccharides linked to either C6 or C20. All PPD and PPT type ginsenosides have different R groups including glucose (Table 2). In the case of an ocotillol type, it is derived from PPT, which has a tetrahydrofuran ring as a side chain [44]. The other type, of oleanolic acid (OA), has an oleanane skeleton as an aglycone of triterpenoid saponins [45,46,47,48,49,50]. Moreover, there are also minor ginsenosides that have a non-polar dehydrated backbone through the loss of glycosidic bonds, such as Rg5, Rk1, and Rk3 [45]. Both ginsenosides Rk1 and Rg5 are dehydrated version at C20 of ginsenoside Rg3, whereas Rk3 is one of the modified PPT types derived from Rh1. In addition, PPD- and PPT-type ginsenosides are typically denoted by Rx (x = b-1, d, g-3…), where x symbolizes the polarity according to the Rf value upon thin-layer chromatography (TLC) in an alphabetical order [51,52]; in the case of the ginsenosides in Table 1, the most polar ginsenoside is Rb1, and Rc is more polar than Rg3.

*Panax ginseng* is known to have various health benefits, which are applied widely, such as improving psychologic function or reducing stress and anxiety. Twenty days of 40 μM/kg of ginsenoside Rh2 administration recovered spatial and non-spatial memory in a sleep-deprivation-induced ICR mouse model by suppressing oxidative stress and antioxidant activity [53]. Additionally, a single dose of 20(S)-Ginsenoside Rg3 with different concentrations showed that 20(S)-Rg3 had concentration-dependent effects on enhancing glucose-induced insulin levels in both HIT-T15 cells and ICR mice [54]. Another study demonstrated that neuroprotective effects of Rh1 and Rg2 (10 μM and 20 μM treated groups) were shown through regulating 6-OHDA-induced ERK1/2 phosphorylation on neuronal cells (SH-SY5Y and PC-12) [55]. Among this range of effects, various studies have paid attention to ginsenosides since it has been revealed that ginsenosides are the active compounds responsible for the anticancer and immunomodulatory effects of *P. ginseng* [30,56,57,58,59]; however, although there is growing interest in the anti-inflammatory effect of ginsenosides, their role in modulating the NF-κB pathway remains poorly understood.

Numerous studies on the functional role of NF-κB in pro-inflammatory responses have been reported. The anti-inflammatory activities of ginsenosides are also of constant interest; however, there are no reviews of the relationship between NF-κB and ginsenosides. Here, we summarize current trends in the anti-inflammatory properties of ginsenosides, especially those targeting the NF-κB pathway. We searched for studies conducted since 2020 in PubMed and Embase that focused on the ginsenoside-derived modulation of NF-κB activity, which eventually leads to the attenuation of inflammatory diseases in both cellular and animal models. As the incidence of chronic and severe inflammatory diseases is increasing, we hope that our efforts can suggest possible directions for the management of these diseases.

## 2. NF-κB and Immune Responses

NF-κB is known as a biomarker in numerous inflammatory diseases, such as inflammatory bowel disease (IBD), atherosclerosis, and COPD [60,61]. The NF-κB-mediated secretion of TNF-α, IL-6, IL-1, and other chemokines induces neutrophil and monocyte recruitment to the inflammatory sites via upregulating biological processes, such as vasodilation or extravasation [62,63,64]. The activation of NF-κB signaling also enhances adaptive immunity insofar that it supports CD4^+^ T cell differentiation in both Th1 and Th17 pro-inflammatory T cells, which are closely linked to the development of autoimmune or inflammatory diseases [65]. The expression of RORγT and RORγ, which are Th17-lineage-specific transcription factors, is known to demand p65 activation.

The atypical NF-κB signaling pathway also regulates lymphoid organ development and adaptive immunity (Figure 2) [66]. In atypical NF-κB signaling, the receptor activators of NF-κB (RANK), B cell activating factor receptor (BAFFR), and lymphotoxin beta receptor (LTβR) are stimulated by the receptor activators of NF-κB ligand (RANKL), B cell activating factor (BAF), and lymphotoxin alpha/beta (LTα1β2), respectively, which subsequently trigger p100 phosphorylation via promoting the enzyme activity of IKKα [67]. The activation of p100 induces the translocation of RelB and p52, which enhances the expression of genes such as granulocyte-macrophage colony-stimulating factor (GM-CSF) for Th17 cell activation [68,69]. The activation of RelB also represses the canonical NF-κB pathway [70]. RelB induces dimmer switch by alternatively binding to p50, replacing p65 [71]. RelB also recruits Sirt1 for histone H3 deacetylation and methylation to induce the epigenetic silencing of the expression of pro-inflammatory genes [72].

## 3. Pharmacological Activities of Ginsenosides on Inflammatory Diseases

Inflammation is initiated by various molecular pathways. Mitogen-activated protein kinase (MAPK), which triggers the phosphorylation-induced activation of AP-1 subunits, activates pro-inflammatory signals that enhance T-cell differentiation and B-cell class switching [73]. The Janus kinase (JAK)-mediated activation of signal transduction and activator of transcription (STAT) also induces the expression of pro-inflammatory agents [74]. Among these pathways, the anti-inflammatory process of *P. ginseng* mainly targets NF-κB activation. The *P. ginseng* saponin fraction with a rich amount of ginsenosides blocks the phosphorylation of IKKα/β, downregulating IκBα degradation and eventually inhibiting the translocation of NF-κB subunits [75]. Interestingly, a mass analysis of the saponin fraction showed that Rb1 accounts for 20% of the total ginsenoside ratio. It has recently been reported that Rb1 attenuated lipopolysaccharide (LPS)-induced kidney injury and dimethyl-benzene-induced ear edema [76]. The mechanism relies on the Rb1-mediated inhibition of TLR4 dimerization, which reduces Myd88 recruitment and continuous NF-κB subunit activation. The OT-type saponin fraction also suppressed LPS-induced inflammation by inhibiting the phosphorylation of IκBα and p65 [77].

### 3.1. Neural Inflammation and Neurodegenerative Diseases

In the central nervous system (CNS), microglia play a key role in innate immunity. Microglia can be either polarized into M1 microglia, which secrete pro-inflammatory cytokines, such as TNF-α, IL-1β, and IL-6, or M2 microglia, which are responsible for neurogenesis and remyelination [78]. Microglia differentiate into M1 microglia under pathological conditions, undergoing pro-inflammatory reactions; however, overactive or uncontrollable microglia activation may result in severe neural inflammation, leading to neurodegenerative diseases. Severe neural inflammation damages the blood–brain barrier (BBB), causing hemorrhagic traits and cerebral edema, which impair neurons [79,80]. Therefore, discovering an effective anti-inflammatory agent that suppresses microglial activation will help cure neurodegenerative disorders. In fact, ginsenoside re-inhibited pro-inflammatory signals in both primary microglia cells and BV2 murine microglia cells induced by LPS [81]. The re-blocked degradation of IκBα suppressed the transcriptional activity of p65.

Stroke is the second most fatal disease following cancer. About 13 million new incidents of stroke are reported annually [82]. Surprisingly, Su et al. showed that ginsenoside Rb1 ameliorates ischemic stroke induced by the occlusion and reperfusion of the cerebral artery in a mouse model [83]. Rb1 attenuates brain injury by improving the tight junction between neurons via zonula occludens-1 (ZO-1) and occludin activation. Microglia activity is inhibited by the facilitating effect of Rb1 on peroxisome proliferator-activated receptor gamma (PPARγ), which inhibits p65 phosphorylation. Ginsenoside Rd has a therapeutic effect on ischemic stroke by enhancing IκBα expression [84]. This inhibits the NF-κB-mediated expression of matrix metalloproteinase (MMP)-9. Since increased MMP-9 expression under acute inflammation impairs the BBB via proteasome activation, Rd successfully protects the BBB.

Ginsenosides not only recover brain damage, but they also cause behavioral improvements. For instance, 20(S)-PPD attenuated depression and improved the sucrose preference of Sprague Dawley rats with chronic unpredictable mild stress (CUMS) [85] in addition to reducing the secretion of *i*NOS, COX-2, IL-1β, IL-6, and TNF-α by suppressing p65 activity, eventually attenuating stress-induced inflammation. Rg1 also recovered the cuprizone (CPZ)-induced populational increase in microglia [86]. Ionized calcium-binding adapter molecule-1 (Iba-1), a microglia biomarker, was reduced in CPZ-induced C57BL/6 mice after Rg1 treatment. The protected axons in the corpus callosum and blocked demyelination lead to behavioral improvements. Interestingly, the decrease in the population of microglia was triggered by a decrease in chemokine ligand 10 (CXCL10) expression mediated by NF-κB. Rg1 blocked the translocation of NF-κB subunits, eventually improving neurodegeneration. Rg1 also showed therapeutic potential in a Huntington’s disease model designed in C57BL/6 mice [87]. The oral administration of Rg1 reversed the apoptosis of striatal neurons caused by the intraperitoneal injection of 3-nitropropionic acid (3-NP). When placed in an open field, the behavior of the 3-NP-injected mice was greatly improved, insofar that the total distance traveled significantly increased. It has been found that Rg1 downregulates the phosphorylation of p65, decreasing the level of TNF-α and IL-1β and blocking the attack of microglia on neuron.

To sum up, ginsenosides modified pro-inflammatory microglial activities by modulating the NF-κB signaling pathway; therefore, they can be used as a possible option for neural disorders.

### 3.2. Myocardial Inflammation and Heart Failure

According to the studies from the American Heart Association in 2013, patients with heart failure in the United States were projected to be 3% of the population by 2030 [88]. Currently, about 64 million people have heart failure worldwide [89]. This life-threatening disease, with a high incidence rate, is also closely associated with inflammatory responses. Heart failure often originates from myocardial infarction preceded by severe myocardial inflammation. Typically, the NF-κB-mediated upregulation of pro-inflammatory cytokines results in the synthesis of adhesion molecules and chemokines (especially C-C motif chemokine ligand 2) in endothelial cells supporting the extravasation of pro-inflammatory leukocytes into the heart, clearing the dead cardiomyocytes and debris to repair the infarct [90]. Unfortunately, a prolonged inflammatory response worsens myocardial infarction by targeting normal cardiomyocytes. This would accompany systolic dysfunction and left ventricular dilatation [91]. Therefore, the major strategies of therapeutic ginsenosides in targeting cardiovascular injuries have been focused on inhibiting the excessive activation of NF-κB signaling in endothelial cells and cardiomyocytes.

Jin et al. used human umbilical vascular endothelial cells (HUVECs) to examine the therapeutic potential of ginsenoside Rh1 on LPS-induced inflammation [92]. LPS treatment dramatically increased the activation of the NF-κB signaling pathway followed by the activation of adhesion molecules, such as vascular cell adhesion protein-1 and intracellular adhesion molecule-1; however, Rh1 treatment reversed the LPS-mediated effects by blocking TLR2 and TLR4 activation. Ginsenoside Rd also protected HUVECs from nicotine-induced inflammatory responses [93]. Rd treatment reduced angiotensin-II (Ang-II) expression, the excessive expression of which causes hypertension, and suppressed the activation of TLR4 and Myd88. In vivo experiments on Sprague Dawley rats with an intraperitoneal injection of nicotine showed consistent results, insofar that Rd attenuated arterial endothelial damage by modulating the NF-κB signaling pathway.

Ginsenoside Rg1 and Rg3 both reduced NLRP3 inflammasome activity in cardiomyocytes [94,95]. Rg1 recovered cardiac function impaired by LPS treatment and suppressed apoptotic signals in neonatal rat cardiomyocytes. Rg3 ameliorated Ang-II-induced myocardial fibrosis and hypertrophy via NLRP3 inflammasome repression. Both ginsenosides ultimately target the NF-κB signaling pathway, Rg1 by suppressing TLR4 expression and Rg3 by blocking the phosphorylation of NF-κB subunits. Another study on Rg3 suggests that it decreases the inflammatory responses in a coronary-artery-ligation-induced myocardial infarction model built in Sprague Dawley rats [96]. Interestingly, Rg3 increased the expression of Sirt1 and RelB. This indicates that Rg3 enhanced the Sirt1/RelB axis, which is attributed to the downregulation of p65 activity through epigenetic silencing. High mobility group box 1 protein (HMGB1), a highly conserved nuclear protein, participates in the defense mechanism under pathogenic conditions [97,98]. Extracellular HMGB1 binds to the receptor for advanced glycation end products (RAGEs) or TLRs, promoting PAMP/DAMP-induced inflammation [99]. Ginsenoside Rh2 blocks the expression of HMGB1 in an H9C2 rat heart myoblast cell line, which was upregulated through oxygen-glucose deprivation [100]. This eventually triggers the inhibition of NF-κB subunit translocation to the nucleus and NLPR3 inflammasome activation.

The above evidence suggests that various ginsenosides could protect endothelial cells, cardiomyocytes, and heart myoblasts from inflammatory responses stimulated by the NF-κB signaling pathway, attenuating heart failure.

### 3.3. Chronic Liver Diseases and Metabolic Disorder

The National Vital Statistics Report showed that about 4.5 million adults in the United States had chronic liver disease in 2017 [101]. Globally, 1.32 million people die from chronic liver diseases, which is the 11th highest mortality rate [102]. Chronic liver diseases include alcoholic liver disease (ALD), non-alcoholic fatty liver disease (NAFLD), infection-induced hepatitis, and autoimmune hepatitis. Since all types of chronic liver diseases are closely linked to chronic inflammation, controlling the activation of immune cells in the liver should be a major strategy for treatment. Kupffer cells are the resident liver macrophages dominating the entire population of macrophages in the body [103]. Kupffer cells can be polarized into either M1 or M2 Kupffer cells, involved in pro- and anti-inflammatory responses, respectively [104]. The imbalance of M1 and M2 Kupffer cells in ALD and NAFLD patients causes severe hepatocyte injury [105]; therefore, ginsenosides with anti-inflammatory properties could be effective for chronic liver diseases.

According to bioinformatics analyses, ginseng is known to have 31 bioactive ingredients that show anti-ALD properties [106]. Among these, KEGG pathway analyses have indicated that Rg1-mediated anti-ALD activity is derived from its inhibitory effects on TLR/NF-κB signaling. Studies have confirmed that alcohol-induced hepatocyte injury in ICR mice is healed by Rg1 treatment. Levels of biomarkers that represent liver damage, such as alanine transaminase (ALT), aspartate transaminase (AST), lactate dehydrogenase (LDH), and alkaline phosphatase (AKP), are also ameliorated by Rg1 treatment. The therapeutic efficacy relies on its suppressive effects on pro-inflammatory cytokines, such as TNF-α, IL-1β, and IL-6, and transforming growth factor-beta1 (TGF-β1), which is responsible for p65 phosphorylation.

In NAFLD patients, excessive fat accumulation in the liver by a high-fat or high-cholesterol diet puts a strain on liver function [107,108,109]. This may trigger insulin tolerance and increase the adipocyte population, resulting in hepatic lipotoxicity. In this stage, lipotoxic hepatocytes release toxic granules that can activate TLRs in Kupffer cells. Subsequently, nonalcoholic steatohepatitis (NASH) can occur with long-term, excessive inflammatory responses in the liver. Since cytokines such as TNF-α, IL-17, and IL-8, secreted from Kupffer cells, induce hepatocyte injury and liver fibrosis, if patients do not receive proper treatment then NASH will go worsen, resulting in liver cirrhosis accompanied by bleeding and ascites, eventually causing hepatocellular carcinoma [110].

20(S)-Rh1 inhibited the phosphorylation of p65, inactivating the downstream NLRP3 inflammasome activation in the liver tissue of C57BL/6 mice induced by a high-fat diet accompanied with a streptozotocin (STZ) injection [111]. Interestingly, it was discovered that Rh1 directly binds to forkhead box O1 (FoxO1) by forming hydrogen bonds. Since FoxO1 synergistically enhances the NF-κB signaling pathway by binding to the insulin response element, the DNA motif of NF-κB target genes, Rh1-derived therapeutic potential on NAFLD is based on FoxO1-modulation-mediated NF-κB inactivation [112].

Lin et al. discovered the fact that Rb2 ameliorates fat accumulation and induces a decease in body weight in C57BL/6 mice fed a high-fat diet [113]. The daily administration of 40 mg/kg of Rb2 blocked adipocyte pyroptosis by inhibiting the NF-κB signaling pathway. Reduced NF-κB signaling downregulated the expression of caspase-1 and IL-1β, as well as NLRP3 inflammasome activity in adipocytes.

The above studies demonstrate that ginsenosides can be applied to chronic liver diseases or metabolic disorders by reversing Kupffer cell polarization and ameliorating lipotoxicity via targeting the NF-κB signaling pathway.

### 3.4. Lung Inflammation and Pulmonary Disorder

Pulmonary inflammation, affecting more than 510 million patients globally, is a major cause of respiratory disorders such as acute respiratory distress syndrome (ARDS), asthma, and COPD [67]. Pulmonary inflammation accumulates neutrophils and macrophages in the alveolar space, interrupting gas exchange [114]. The TNF-α-induced necrosis of alveolar epithelial cells also results in pulmonary edema via the release of cellular fluid into the airway [115,116]. Moreover, it has been known that individuals with asthma or COPD exhibit the sustained activation of NF-κB signaling in the lung epithelium [117]. Thus, agents that can ameliorate NF-κB signaling are a possible drug option for inflammatory lung diseases.

Cigarette smoke is the primary cause of COPD. According to Li et al., when primary airway basal cells from healthy donors are exposed to cigarette smoke, the activation of the NF-κB signaling pathway is induced [118]; however, ginsenoside Rb3 treatment alleviates the cigarette-smoke-induced activation of p65 and subsequent secretion of IL-6, IL-8, and IL-1β. Their study also demonstrated that Rb3 downregulated cancer-like features such as endothelial–mesenchymal transition (EMT) or hyperplasia induced by cigarette smoke exposure.

Ginsenoside Rb1 deactivates the NLRP3 inflammasome, which participates in IL-1β maturation via caspase-1 activation [119]. IL-1β is a key component involved in the progress of pulmonary fibrosis. Rb1 protects C57BL/6 mice from pulmonary fibrosis induced by bleomycin, which comprises pulmonary edema accompanied by the infiltration of immune cells. Rb1 reverses the immune cell population in lung tissue by inhibiting IκBα degradation, which leads to the inactivation of the NLRP3 inflammasome and IL-1β. Rb1 also downregulates inflammatory responses in an LPS-treated WI-38 human lung fibroblast cell line by targeting NF-κB signaling [120]. Specifically, Rb1 suppresses miR-222 activation, which is responsible for NF-κB activation [121]. The effect of Rb1 was confirmed by the overexpression of miR-222. The overexpression of miR-222 downregulates the therapeutic efficacy of Rb1 in targeting LPS-induced injury in lung fibroblasts. This indicates that Rb1-mediated NF-κB modulation is closely related to its regulatory effect on miR-222 expression.

A daily oral injection of 30 mg/kg of Rg1 also alleviates acute pulmonary inflammation (ALI) induced via the intratracheal administration of LPS [122]. Rg1 ameliorated the phosphorylation of p65 in inflammatory lesions, decreasing the thickening of the alveolar wall. Interestingly, Rg1 treatment activated the formation of autophagosomes by enhancing the transition of microtubule-associated protein 1 light chain 3 (LC3)-I to LC3-II, which induces the activation of Nrf2. Since it has been known that Nrf2 expression suppresses p65 phosphorylation, the authors concluded that the Rg1-mediated anti-inflammatory response is derived from its supportive effect on autophagy, which leads to Nrf2 activation and continuous NF-κB deactivation [123,124].

In summary, ginsenosides can also be applied as drugs against inflammatory pulmonary diseases since ginsenosides can successfully ameliorate immune cell infiltration to the alveolar space, lung edema, the thickening of the alveolar wall, and alveolar cell injury by regulating the NF-κB signaling pathway.

### 3.5. Kidney Inflammation and Nephropathy

Kidney inflammation, which accounts for 40% of kidney disease, affects kidney function and its ability to filter blood as well as excrete waste through urine. Diabetic nephropathy (DN) is one of the leading causes of chronic kidney disease, especially in developed countries [125]. In the United States, about one in three patients with diabetes has DN. This is a common complication of type 1 and type 2 diabetes, and high blood glucose levels in patients with diabetes can damage the small blood vessels in the kidneys during filtration. Recent studies have suggested that NF-κB could contribute to the inflammatory progress in DN. High glucose levels in the blood can activate NF-κB in renal cells, leading to the production of pro-inflammatory cytokines, chemokines, and adhesion molecules [126].

The levels of TNF-α and IL-1β were found to be regulated by 20(R)-ginsenoside Rg3 in a DN mouse model. To induce DN, a high-fat diet (HFD) was continuously provided and STZ (100 mg/kg) was injected. NF-κB was overexpressed in the HFD/STZ group compared to the normal group, whereas reduced NF-κB expression was observed in the 20(R)-Rg3 treated group. Moreover, the increased expression of p-NF-κB, p-IκBα, and p-IKKβ was downregulated by 20(R)-Rg3 at the protein level [127]. This study also demonstrated that 20(R)-Rg3 can inhibit the advanced glycation end products that trigger ROS and cause oxidative stress, leading to the recovery of histopathological injuries in DN.

The protective effect of ginsenoside Rg5 has been significantly shown in HFD/STZ-induced diabetic mice. The expression of IL-6, TNF-α, and IL-18, the main inflammatory cytokines, are downregulated in renal tissues and sera following high doses of Rg5. When NLRP3 is activated an inflammasome complex is formed, which produces and secretes inflammatory cytokines—IL-1β. A Western blot analysis found that elevated levels of NLRP3, pro-IL-1β, and IL-1β were reduced in the kidneys of the DN group by Rg5. In addition, Rg5 significantly ameliorates the level of NF-κB p65 expression, which illustrates the fact that Rg5 plays a protective role in DN kidneys by modulating the NF-κB pathway [128].

Using C57BL/6 mice with contrast-induced nephropathy (CIN), researchers have demonstrated that ginsenoside Rb1 (GRb1), a key active component of ginseng, dose-dependently suppresses the expression of IL-6, IL-1β, and TNF-α. Moreover, high mobility group box 1 (HMGB1), an extracellular inflammatory cytokine involved in transcriptional regulation [129], is known to interact with TLR4. The expression of p-IκBα, nuclear p65, HMGB1, and TLR4 was inhibited by GRb1 in a CIN mouse model. Specifically, under the overexpression of HMGB1, GRb1 can prevent the inflammatory response by inhibiting levels of p-IκBα, nuclear p65, and TLR4 in iopromide (IOP)-treated NRK-52E rat renal epithelial cells. This indicates that GRb1 has an impact on recovering renal tubular epithelial cell injury by targeting HMGB1 via the NF-κB signaling pathway [130].

Overall, these results imply that these ginsenosides can be applied in diabetic kidney injury by blocking the secretion of inflammatory cytokines in addition to alleviation of oxidative stress and apoptosis via mediation of the NF-κB signaling pathway.

### 3.6. Colitis and Digestive Disorder

Colitis refers to inflammation of the colon, as well as a type of digestive disorder. A range of causes can attack the lining of the colon, including dietary factors, infections, and autoimmune disorders [131]. Evidence indicates a relationship between colitis and obesity, suggesting that obesity is a critical factor for developing ulcerative colitis (UC) [132]. Numerous studies report that the NF-κB pathway plays a critical role in controlling the release of cytokines in UC and contributes to the immune response in the intestinal tract of UC. For example, when the NLRP3 inflammasome is activated it triggers the activation of caspase-1 as well as the production of IL-1β [133].

Recently, gut microbiota are some of the hot keywords related to intestinal health. The gut microbiota refer to the numerous bacteria living in the human gut, forming a complex ecosystem. Gut bacteria regulate a host’s immune system by modulating epithelial mucosa and inflammatory cytokine production [134]. For instance, *Lactobacillus fermentum* induces the production of anti-inflammatory cytokines such as IL-10 [135]. On the other hand, gut bacteria that degrade the mucus layer and induce a dysregulated immune response trigger the uncontrollable activation of NF-κB signaling, resulting in serious inflammatory diseases [136]. Thus, the imbalance in the gut microbiota population, also called microbial dysbiosis, must be restored to its original state to cure inflammatory diseases.

Chen et al. examined the role of Rk3 in obesity-induced colitis [137]. Rk3 diminished the expression of TNF-α, IL-1β, and IL-6 in serum and colon tissues in HFD-induced obese mice. The mRNA levels of markers in macrophage infiltration (MCP-1 and F4/80) and oxidative stress (NADPH and STAMP-2) were also estimated. The indices were downregulated by Rk3 in a dose-dependent manner. HFD induced an increase in the secretion of LPS in the plasma, which triggers the activation of TLR4 and its downstream signaling, whereas Rk3 attenuated the TLR4/MyD88 pathway cascade and enhanced IκB expression, factors that inhibit the nuclear translocation of NF-κ*B*. Furthermore, microbial dysbiosis was ameliorated by Rk3 via its suppressive effect on pro-inflammatory bacteria, such as *Bacteroidete*.

Ginsenoside Rc (Rc) inhibits the expression of IL-1β, TNF-α, and IL-6, as well as the nuclear translocation of NF-κ*B* in LPS-induced human intestinal epithelial LS174T cells. This effect was also investigated at an in vivo level in a UC model induced by dextran sulfate sodium (DSS). The expression of IL-1β, IL-6, COX-2, TNF-α, and ICAM was repressed by Rc. Specifically, the researchers focused on the function of the farnesoid x receptor (FXR), a nuclear receptor activated by bile acids, in the inflammatory response of the intestine [138]. The mRNA level of FXR as well as ZO-1 and claudin-1 was enhanced with Rc treatment in DSS-induced mice. The function of FXR was elucidated by the cytokine level, which was reversed in DSS-induced FXR^−/−^ mice. Their results indicate that Rc suppressed the inflammatory response and recovered damage to intestinal barriers [139].

In acute radiation proctitis (ARP), the effect of ginsenoside Rg3 (GRg3) was investigated in an ARP rat model. Rg3 plays an active role in anti-inflammation dose-dependently by restricting the expression of TLR4, MyD88, NF-κB p65, IL-10, and IL-1β at the mRNA and protein levels. This indicates that GRg3 regulates NF-κB activation via MyD88-dependent TLR4. In addition, imbalances in the composition of gut microbiota were improved after treatment with GRg3, insofar that the diminished population of immunomodulatory microbiota, such as *Lactobacillus* and *Ruminococcus*, was increased while the population of pro-inflammatory bacteria, such as *Alloprevotella*, was decreased [140].

In sum, ginsenosides have a potential therapeutic effect on colitis by regulating as well as protecting the intestinal mucosa and altering the composition of microbiota.

### 3.7. Bone Disorders

Bone is a living growing tissue that plays an important role in supporting the body and protecting internal organs. If abnormal bone function occurs, bone fractures can be induced through decreased bone microstructure and bone strength. Activated T and B cells promote bone loss by secreting cytokines, including TNF-α, RANKL, and IL-17A [141]. Common bone disorders include osteoporosis (OP) and osteoarthritis (OA).

In OP, fractures occur easily due to weakened bone strength, which mainly occurs in the spine and wrist bones but may affect the whole body. When the amount of bone removed and decomposed without the proper formation of new bones becomes excessive, a lack of bone mass ensues [142]. OP can lead to complications such as infections and blood clots, which can lead to potentially life-threatening states, especially in older adults [143].

OA is a degenerative joint disease that impacts cartilage, which is the protective tissue that covers the ends of bones in a joint. This disease can affect any joint, but mostly affects the knees and is prevalent globally in those older than 60 years old [144]. Chronic inflammation in the joint can also contribute to the progression of OA. Inflammation can cause damage to the remaining cartilage, leading to further breakdown and the worsening of symptoms [145].

The therapeutic effects of Rb1 are shown in both OP and OA. First, in a dexamethasone (DEX)-induced rat model of OP, the NF-κB p65 protein expression was inhibited. According to recent studies, an aryl hydrocarbon receptor (AHR), related to immune responses, triggers the growth and differentiation of osteoblasts, which leads to the recovery of bone injury. This study demonstrated that, in OP rats, AHR elevated the level of proline and arginine-rich end leucine-rich repeat protein (PRELP), known as a factor involved in the inhibition of osteoclastogenesis [146]. Subsequently, NF-κB p65 protein expression was reduced under the upregulation of PRELP. Thus, GRb1 can promote the osteogenic differentiation function by blocking NF-κB via the AHR/PRELP signaling cascade [147].

Rb1 was also examined in a rabbit knee OA model. The increased expression of TNF-α, caspase-3, and BAX was reversed in the Rb1-treated group. Rb1 also diminished PGE2 production and MMP expression by targeting p-Akt/Akt, p-P65/NF-κB, and p-P38/P38 proteins in a dose-dependent manner, which was confirmed through macroscopic images that showed recovered cartilage injuries. These results indicate that GRb1 can restore an impaired articular cartilage area by downregulating the level of MMPs through the suppression of apoptosis and the NF-κB signaling pathway [148].

Lastly, the impact of Rb3 on alveolar bone resorption was explored by Sun et al. Enhanced IL-8, IL-6, and IL-1β levels induced by *P. gingivalis* LPS were shrunk by Rb3 under non-cytotoxic conditions. Rb3 also attenuated p38 MAPK, AKT, and NF-κB activation in human periodontal ligament cells (HPLCs). In particular, Rb3 significantly reduced TLR2 mRNA expression. This tendency was demonstrated in an experimental periodontitis rat model. Osteoclast genesis and bone loss were diminished after Rb3 treatment. These data support the protective effect of Rb3 against alveolar bone resorption with the deactivation of the immune response [149].

In conclusion, these ginsenosides can be considered therapeutic agents of bone disorders by differentiating osteoblasts and through the downregulation of NF-κB signaling pathways.

## 4. NF-κB and Cancer

Cancer is a major public health concern with a high mortality rate. Approximately two million new cases of cancer and 0.6 million deaths are reported annually in the United States [150]. Although there are numerous efforts to prevent and mitigate cancer, not every mechanism of cancer is known, insofar that there are too many factors involved in the development and spread of cancer. The properties of tumors associated with invasion, migration, EMT, and immune resistance make it difficult to remove tumors completely [151,152,153]. Interestingly, many studies demonstrate that NF-κB signaling participates in the progression of cancer, indicating that NF-κB can be used as a key target in the development of anticancer drugs (Figure 3) [154,155].

The invasion and migration of cancer cells result in tumor metastasis, making treatment with surgical resection difficult. Malignant tumor cells lose cell–cell adhesion capacity and detach from the primary tumor mass [156]. The dissociated tumor cells gain motility and the ability to migrate by building blood vessels via a process called angiogenesis [157]. Once the synthesized blood vessels link with adjacent blood vessels, the detached cells can be transported via the circulatory system and metastasize to other organs. The dissociation of malignant cancer cells demands the activity of MMPs [158]. MMPs, especially MMP-2 and MMP-9, degrade gelatin and type IV collagen, collapsing the structure of the extracellular matrix (ECM) [159]. The isolated cancer cells gain a mesenchymal feature with higher invasiveness by expressing transcription factors such as zinc finger protein SNAI1 (Snail), zinc finger protein SNAI2 (Slug), and twist-related protein 1 (TWIST1), which suppress epithelial biomarkers [160]. This change in the characteristics of cancer cells is called EMT. In fact, NF-κB is a positive regulator of migration, invasion, and EMT by activating the transcription of genes encoding MMPs, Snail, Slug, and TWIST1 [161,162]. It is also known that NF-κB upregulated by the interaction of TNF-α and TNFR in cancer cells stabilizes EMT markers [163]. Therefore, the inhibition of the NF-κB pathway in malignant tumors can ameliorate metastasis by repressing the migration and invasion of cancer cells.

Oncogenic receptors, especially receptor tyrosine kinases such as epidermal growth factor (EGFR), proliferate the activation of protein kinase B (Akt) [164]. Akt activation triggers IKKα/β, which subsequently results in the translocation of NF-κB [165]. NF-κB is known to bind to the promoter region of the programmed death-ligand 1 (PD-L1) coding gene, inducing the overexpression of PD-L1 [166]. PD-L1 is presented to the membrane of cancer cells interacting with programmed cell death protein 1 (PD-1) in cytotoxic T lymphocytes. The interaction between PD-L1 and PD-1 interrupts the regulatory role of T lymphocytes in killing tumor cells. Notably, pro-inflammatory processes initiated by extrinsic interferon-gamma (IFN-γ) or TNF-α also facilitate the expression of PD-L1 by promoting the nuclear translocation of NF-κB subunits [167]. Therefore, agents that can inhibit the NF-κB signaling pathway may overcome the avoidance mechanism of cancer cells. Several studies have shown that tyrosine kinase inhibitors, such as gefitinib, downregulate the level of PD-L1 by modulating the NF-κB pathway.

There are thousands of studies on the anticancer capacity of ginsenosides [168,169,170,171]. Most studies focus on their anti-proliferative and pro-apoptotic roles against cancer cells; however, the severity of the disease is derived from metastatic properties and therapy resistance. Therefore, therapeutic agents that regulate NF-κB, which is a key modulator of cancer migration and invasion, are in the limelight as a possible cure for cancer. Recent studies also cover the anticancer activity of ginsenosides, which target the NF-κB pathway. Therefore, we wish to introduce studies that demonstrate ginsenosides as an effective cancer therapy that targets NF-κB-mediated metastatic capacity.

## 5. Anticancer Effects of Ginsenosides via NF-κB Modulation

Recent studies on Rh1, Rk1, and Rg5 have shown that ginsenosides can block the translocation of NF-κB subunits, which reduces the NF-κB-mediated expression of MMPs [172,173]. Since MMPs are involved in tumor cell migration and invasion, Rh1 suppressed the migration of the MDA-MB-231 human breast cancer cell line while Rk1 and Rg5 reduced the invasion of the A549 human lung adenocarcinoma cell line. Rh1 induces mitochondrial ROS, negatively regulating the phosphorylation of STAT3. Lowering STAT3 phosphorylation reduces the phosphorylation of p65 and inhibits its function as a transcription activator. Meanwhile, Rk1 and Rg5 can control EMT in the lung cancer cell line that is upregulated by TGF-β1 via modulating the expression of E-cadherin and vimentin.

The therapeutic activity of Rk1 was also evaluated in an A549 xenograft mouse model [174]. Rk1 treatment induced a loss in tumor size without the modulation of white blood cell population and organ damage, which was seen in the gefitinib-treated positive control group. Rk1 also suppressed the expression of PD-L1 by inhibiting the NF-κB signaling pathway. Since PD-L1 binds to PD-1 from lymphocytes and helps cancer cells to avoid immune responses, Rk1 can also be used as a possible immune checkpoint inhibitor with fewer side effects.

Ginsenoside Rh2 is well known for its anticancer properties [175]. Nearly 200 studies on its therapeutic potential against cancer have been reported. Revealing the specific mechanisms of anticancer effects is a major concern for researchers. Wang et al. found that 20(S)-Rh2 directly binds to annexin A2 (Anxa2) [176]. Anxa2 interacts with p50, supporting the NF-κB-mediated expression of Snail, Slug, TWIST1, and MMPs, which are responsible for EMT. Interestingly, the physical interaction between 20(S)-Rh2 and Anxa2 blocked the binding of Anxa2 and p50, ameliorating the signaling activity of NF-κB. Therefore, the migration as well as invasion of both the MDA-MB-231 and MCF-7 human breast cancer cell lines were inhibited by 20(S)-Rh2 treatment in a dose-dependent manner. Li et al. revealed that Rh2 inhibits the proliferation of U20S cells by targeting the phosphatidylinositol 3-kinase (PI3K)/Akt/mechanistic target of rapamycin (mTOR) axis [177]. Since mTOR induces the phosphorylation of IKKα/β, the Rh2-mediated deactivation of the PI3K/Akt/mTOR pathway depresses NF-κB activation [178]. This results in the impairment of migration in U20S cells.

## 6. Conclusions and Perspectives

The therapeutic efficacy of *P. ginseng* has been studied for thousands of years; however, research on its modulatory effect on the NF-κB signaling pathway does not have a long history. The first paper reported, published in 2001, deals with *P. ginseng*’s antitumor activity, derived from its inhibitory effects on the NF-κB and MAPK signaling pathways examined in human leukemia cells [179]. Recently, it has been found that black ginseng, which is achieved from a repetitive steam–dry cycle of *P. ginseng*, ameliorated p65 phosphorylation, reducing liver inflammation in a mouse NASH model [180]. Meanwhile, a ginsenoside mixture, obtained from ginseng stem and leaf, protected hepatocytes from apoptotic signals and inflammatory damage initiated by the upregulation of NF-κB signaling [181].

In this review, we narrowed down the effects of each single ginsenoside rich in *P. ginseng*. This review presents recent studies that have demonstrated the anti-inflammatory and anticancer activities of ginsenosides, especially in targeting the NF-κB pathway (summarized in Figure 4). Ginsenosides ameliorate the polarization of innate immune cells, such as microglia or Kupffer cells, by attenuating the expression of pro-inflammatory cytokines via NF-κB inhibition. Rg1 reduces TNF-α- and IL-1β-positive microglia, recovering the population of striatal neurons and behavior capacities [87]. The inhibitory effects on the NF-κB activation of ginsenosides are acquired by targeting different steps of the NF-κB signaling pathway. For instance, Rh1 targets the initial step of the canonical NF-κB signaling pathway by suppressing the expression of TLR2 and TLR4, ameliorating the inflammatory process in HUVEC cells [92]. Rg3 upregulates the non-canonical NF-κB pathway by inducing RelB activation, which induces the epigenetic silencing of p65 target genes [96]. In the case of Rh2, it directly interacts with Anxa2, inhibiting p50 activity, which induces the expression of Snail, Slug, and TWIST1 [176]. To sum up, ginsenosides can ameliorate severe inflammatory diseases and cancer by regulating the NF-κB signaling pathway.

The effects of ginsenosides on chronic diseases are quite promising, since small doses of ginsenosides showed significant therapeutic effects. As we can see in Table 3 and Table 4, less than 100 mg/kg of pure ginsenosides attenuated diseases in animal model studies. The interesting point is that ginsenosides have therapeutic effects comparable to those of commercially available positive control drugs. Metformin is a drug for type 2 diabetes that also has significant anti-inflammatory effects on monocytes [182]. Surprisingly, it was confirmed that Rh1 has the effect of inhibiting the expression of TNF-α and IL-1β, comparable to metformin [92]. The administration of 100 μg/kg of Rb1 also exerted stronger inhibitory effects on the expression of pro-inflammatory cytokines compared to 5 mg/kg of diclofenac, which is one of the well known NSAIDs for stroke and arthritis [148]. Thus, it raises expectation that ginsenosides can replace drugs sold in pharmacies.

Ginsenosides are bioactive compounds abundant in *Panax ginseng* that exert anti-inflammatory, antimicrobial, antidiabetic, and anticancer activities. The global ginseng market size is estimated to be USD 2084 million, and the development of extraction techniques allows for the mass production of ginsenosides, such as Rg1, Re, Rh2, and Rg3, in tens of kilograms [33]; however, there are several steps that are necessary in order to overcome challenges in using ginsenosides as possible medicines. Although there are about 4000 research papers evaluating the toxicity of ginsenosides, most of these papers only focus on the misuse or abuse of ginseng at the in vivo level, not covering the toxicological data of patients with certain inflammatory diseases [183]. Moreover, not many clinical trials with ginsenosides have been conducted with the purpose of considering therapeutic efficacies [184]. Ginsenoside Rg3 was administered to non-small-cell lung cancer patients, improving survival rates, but more clinical trials should be conducted to build up enough data [185]. The reason for the small number of clinical trials may be associated with the lack of suitable carriers delivering ginsenosides to appropriate tissues. Since most ginsenosides, such as Rb2, exert a poor absorption rate and rapid tissue distribution, an optimal biocarrier should be developed for a long-lasting effect of ginsenosides on the human body [186,187]. Promisingly, there are growing efforts to build biocompatible carrier packaging ginsenosides [188,189]; for instance, an Rg3-loaded hydrogel, made with mPEG-b-PLGA polymers for increased delivery efficiency, was developed to target perianal ulcers in a rat model [190].

In conclusion, based on this review, we hope that more researchers understand the specific mechanisms of the therapeutic efficacy of ginsenosides, and that more clinical trials are conducted to design commercialized drugs with ginsenosides as active compounds.

## Figures and Tables

**Figure 1 ijms-24-06119-f001:**
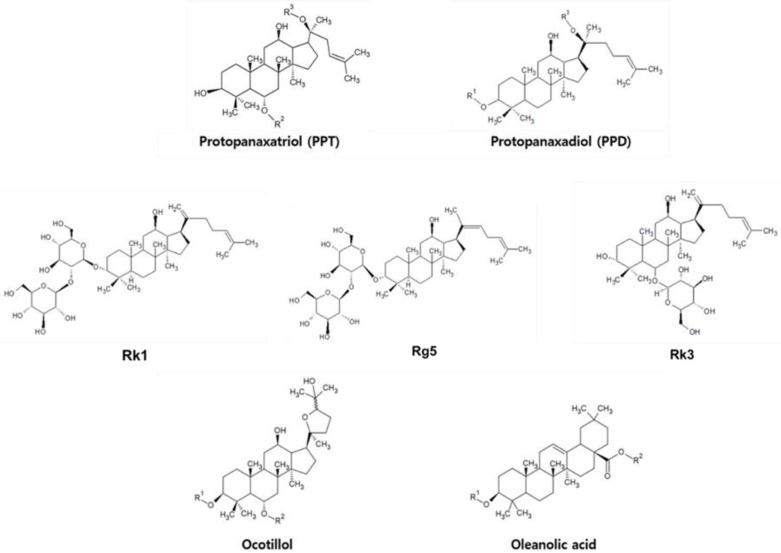
Chemical structures of ginsenosides with derivatization positions indicated by R groups (R^1^ (C3), R^2^ (C6), and R^3^ (C20)).

**Figure 2 ijms-24-06119-f002:**
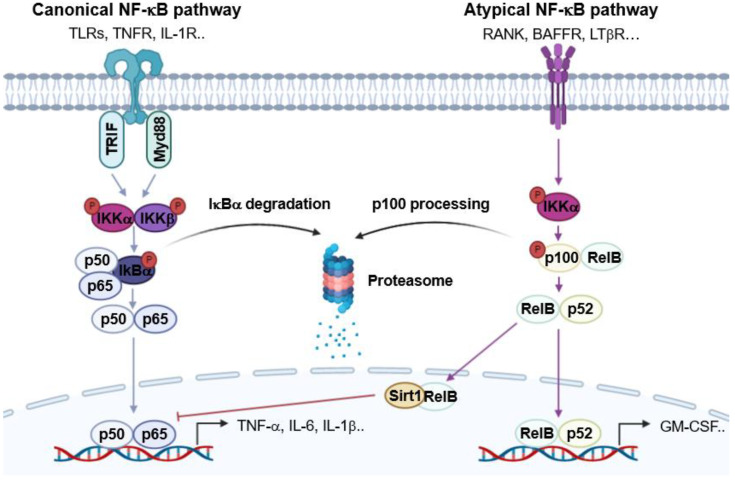
Graphical analysis of the NF-κB signaling pathway in the inflammatory response.

**Figure 3 ijms-24-06119-f003:**
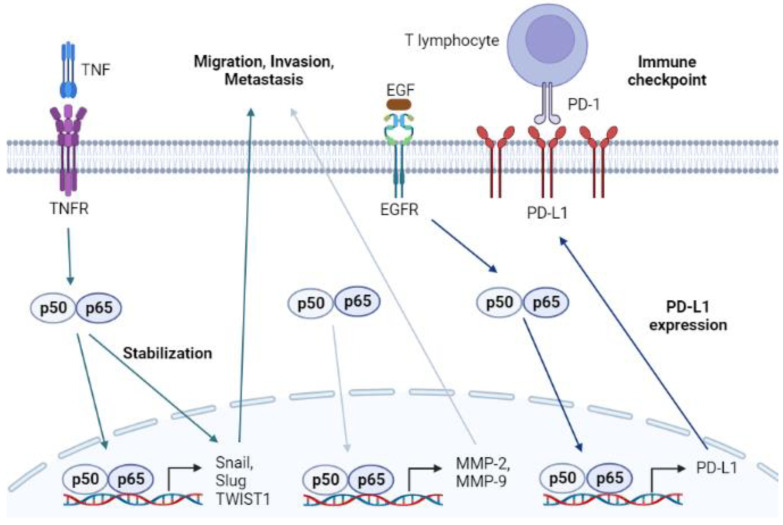
Graphical analysis of the NF-κB signaling pathway in cancer.

**Figure 4 ijms-24-06119-f004:**
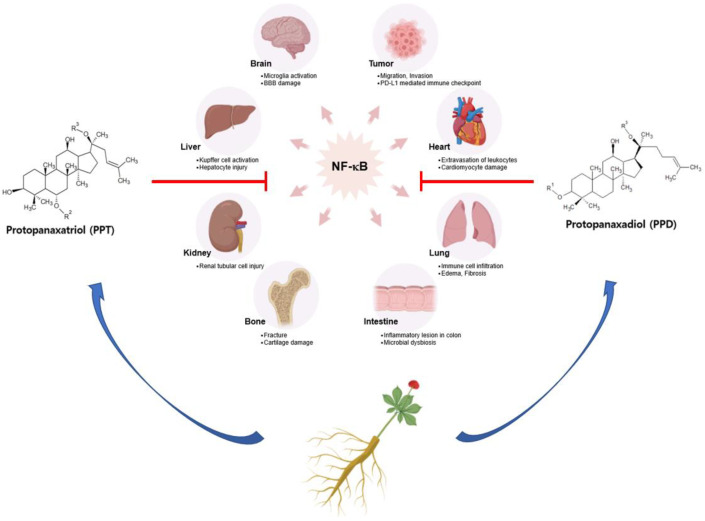
Overall scheme of the immunopharmacological effects of ginsenosides targeting the NF-κB signaling pathway.

**Table 1 ijms-24-06119-t001:** PPD- and PPT-type ginsenosides stated in this paper.

PPD Type	PPT Type
Rb1, Rd, Rg3, 20(R)-Rg3, Rh2, Rb3, 20(S)-Rh2, Rk1, Rg5, and Rc	Rg1, Rh1, Rk3, R1

**Table 2 ijms-24-06119-t002:** The structure of PPD- and PPT-type ginsenosides with their R groups.

**PPT**	**Ginsenoside**	**R^2^**	**R^3^**
	Rg1	OGlc	Glc
	Rh1	OGlc	H
	R1	Glc(2 → 1)Xyl	Glc
**PPD**	**Ginsenoside**	**R^1^**	**R^3^**
	Rb1	Glc(2 → 1)Glc	Glc(6 → 1)Glc
	Rd	Glc(2 → 1)Glc	Glc
	Rg3	Glc(2 → 1)Glc	H
	20(R)-Rg3	Glc(2 → 1)Glc	H
	Rh2	Glc	H
	20(S)-Rh2	Glc	H
	Rb3	Glc(2 → 1)Glc	Glc(6 → 1)Xyl
	Rc	Glc(2 → 1)Glc	Glc(6 → 1)Araf

**Table 3 ijms-24-06119-t003:** Therapeutic activities of ginsenosides on inflammatory disease models.

Inflammation Site	Test Type	Dose	Type	Molecular Mechanism	Ref.
Neurodegenerative disorder	In vivo (Middle cerebral artery occlusion-/reperfusion-induced mouse ischemic stroke model.) (Strain not mentioned.)	Not mentioned	Rb1	Rb1 reduces the phosphorylation of p65 via activating PPARγ p65 inhibition reversed the expression of ZO-1 and occludin, attenuating brain injury	[83]
	In vivo (Middle cerebral artery occlusion-/reperfusion-induced Sprague Dawley rat ischemic stroke model.)	30 mg/kg	Rd	Rd downregulates proteasome activity, which degrades IκBα The Rd-mediated upregulation of cellular IκBα represses the activation of p50 and p65, subsequently inhibiting MMP-9 expression Via reducing MMP-9 expression, Rd protects the BBB from ischemic stroke	[84]
	In vivo (CUMS-induced Sprague Dawley rat depression model.)	20, 40 μM/kg	20(S)-protopanaxadiol	20(S)-PPD upregulates the level of Sirt1 in the rat hippocampus, inducing p65 inactivation 20(S)-PPD blocks apoptosis in the rat hippocampus Through p65 inactivation, 20(S)-PPD recovers depression-like behavior	[85]
	In vitro (LPS-treated CTX TNA2 cell line.) In vivo (CPZ-induced C57BL/6N mouse demyelination model.)	40 μM 5, 10, and 20 mg/kg	Rg1	Rg1-mediated NF-κB inhibition induces reduced CXCL10 expression Reduction in CXCL10 expression in the brain downregulated the migration and pro-inflammatory capacity of microglia Rg1 protects myelin from the phagocytosis of microglia	[86]
	In vivo (3-NP-induced C57BL/6 mouse Huntington’s disease model.)	10, 20, and 40 mg/kg	Rg1	Rg1 downregulates MAPKs and the NF-κB signaling pathway, reducing the expression of IL-1β and TNF-α Rg1-mediated inactivation of NF-κB underlies its inhibitory effect on microglia Rg1 exerts protective activity on BBB and striatal neuron against 3-NP	[87]
Myocardial injury	In vitro (LPS-treated HUVEC cell line.) In vivo (LPS-induced C57BL/6 mouse endothelial cell injury model.)	25, 50 μM 5 mg/kg	Rh1	Rh1 blocks TLR2/TLR4 activation, modulating pro-inflammatory processes, including the NF-κB signaling pathway Rh1 reverses apoptotic signals and the cell cycle arrest of HUVEC cells Rh1-mediated anti-inflammatory response induces a reduction in cytokines and adhesion molecules	[92]
	In vitro (Nicotine-treated HUVEC cell line.) In vivo (Nicotine-induced Sprague Dawley rat endothelial cell injury model.)	7.5, 15, and 30 μM 25, 50 mg/kg	Rd	Rd reversed TLR4 activation, inducing a decrease in Myd88 expression and p65 phosphorylation Rd refrains nicotine-induced HUVEC-monocyte adhesion by alleviating the expression of adhesion molecules Rd protected aortic endothelial cell injury from nicotine treatment	[93]
	Ex vivo (LPS-treated neonatal rat cardiomyocytes.) In vivo (LPS-induced C57BL/6J mouse septic cardiac dysfunction model.)	20 μM Not mentioned	Rg1	Via TLR4 inhibition, Rg1 suppresses NF-κB signaling and subsequent NLRP3 inflammasome activation Rg1 promotes the activation of anti-apoptotic proteins, attenuating apoptosis in cardiomyocytes Rg1 administration recovers heart function in myocardial injury models	[94]
	In vitro (Angiotensin-II-treated AC16 and HCM cell line.) In vivo (Transverse aortic constriction-induced Sprague Dawley rat myocardial hypertrophy model.)	10, 20, and 40 μM 30 mg/kg	Rg3	Rg3 boosts Sirt1 function in attenuating the NF-κB signaling pathway Reduction in the NF-κB signal triggers lower levels of NLRP3 inflammasome in cardiomyocytes Rg3 also exerts antioxidant effect in cardiomyocytes Rg3 attenuated myocardial hypertrophy in both in vitro and in vivo models	[95]
	In vivo (Coronary-artery-ligation-induced Sprague Dawley rat myocardial infarction model.)	30 mg/kg	Rg3	Rg3 promotes Sirt1 and RelB to inhibit the transcription of pro-inflammatory cytokines induced by p65 The anti-inflammatory effect of Rg3 improves cardiac function in animal myocardial infarction models	[96]
	In vitro (Hypoxia-treated H9C2 cell line.)	2 μM	Rh2	Rh2 ameliorates HMGB1 activation, downregulating NF-κB subunit translocation and activation in hypoxic conditions	[100]
Metabolic disorder	In vivo (ICR mouse alcoholic liver damage model.)	10, 40 mg/kg	Rg1	Through the alleviation of p65 phosphorylation, Rg1 downregulates TNF-α, IL-1β, IL-6, and TGF-β1 The reduced inflammatory response heals liver damage by attenuating ALT, AST, LDH, and AKP levels enhanced by alcohol administration	[106]
	In vivo (STZ-induced C57BL/6 mouse liver injury model.)	5, 10 mg/kg	20(S)-Rh1	The binding of 20(S)-Rh1 and FoxO1 interferes with NLRP3 inflammasome activation via reducing NF-κB signaling activity	[111]
	In vitro (TNF-α-treated 3T3-L1 preadipocyte cell line.) In vivo (HFD-induced C57BL/6 mouse obesity model.)	5 ng/mL 40 mg/kg	Rb2	Rb2 recovered body weight increase, fat accumulation, and insulin resistance Rb2-mediated inhibitory effect on adipocyte pyroptosis comes from the blockading of caspase-1, the NLRP3 inflammasome, and IL-1β Rb2 blocked the phosphorylation of IκBα and p65	[113]
Pulmonary inflammation	Ex vivo (CSE-treated human airway basal cells.)	10, 20, 50 μM	Rb3	The CSE-induced activation of MAPKs and NF-κB was ameliorated by Rb3 Rb3-mediated NF-κB suppression blocks the secretion of IL-1β, IL-6, and IL-8 Rb3 also targets TROP2 expression, inhibiting airway basal cells to have tumor-like features, including EMT	[118]
	In vitro (LPS-treated THP-1 and MRC-5 cell line.) In vivo (BLM-induced C57BL/6 mouse pulmonary fibrosis model.)	20 μM 20 mg/kg	Rb1	Rb1 attenuates the NLRP3-inflammasome-mediated maturation of IL-1β via targeting the NF-κB pathway Rb1 administration reduces inflammatory lesions and fibrosis in a mouse lung treated with BLM	[119]
	In vitro (LPS-treated WI-38 cell line.)	20, 30 μM	Rb1	Via negatively downregulating miR-222, Rb1 reduces NF-κB signaling Rb1-mediated miR-222 inactivation eventually leads to improvement in lung injury caused by LPS treatment	[120]
	In vitro (LPS-treated MLE-12 cell line.) In vivo (LPS-induced C57BL/6 mouse ALI model.)	Not mentioned 30 mg/kg	Rg1	Rg1 induces autophagy, which continuously triggers Nrf2 expression Elevated levels of Nrf2 interrupt p65 phosphorylation, inducing anti-inflammatory effects The Rg1-mediated anti-inflammatory response protects the lung epithelium from immune cell infiltration	[122]
Kidney inflammation	In vivo (HFD-induced C57BL/6 mouse T2D model.)	30, 60 mg/kg	20(R)-Rg3	20(R)-Rg3 blocks apoptosis by regulating the expression of BAX, caspase 8, Bcl-2 and Bcl-XL. 20(R)-Rg3 alleviated oxidative stress and reduces TNF-α, IL-1β expression via MAPK/NF-κB signaling pathway	[127]
	In vitro (Iopromide-treated NRK-52E cell line.) In vivo (CIN C57BL/6 mouse nepropathy model.)	Not mentioned 20, 70 mg/kg	Rb1-nanoparticle (PEG/PLGA)	GRb1-mediated inhibition of the apoptosis of tubular epithelial cells GRb1-mediated anti-inflammatory response by blocking IL-6, IL-1β, HMGB1, and TNF-α secretion through the deactivation of NF-κB	[130]
	In vivo (HFD/STZ-induced C57BL/6 mouse diabetes model.)	30, 60 mg/kg	Rg5	Rg5 alleviates the effects of NAD(P)H oxidase and oxidative stress Rg5-mediated suppression of apoptosis in renal tubular and cytokines production (IL-18, IL-1β and TNF-α) Rg5 blocks NF-κB/MAPK signaling pathway	[128]
Colitis	In vivo (Obesity-induced C57BL/6 mouse colitis model.)	30, 60 mg/kg	Rk3	Rk3 reduces the levels of IL-6, IL-1β, and TNF-α in serum by inhibiting the TLR4/NF-κB signaling pathway Rk3 diminishes macrophage infiltration and attenuates oxidative stress	[137]
	In vitro (LPS-induced LS174T cell line.) In vivo (DSS-induced C57BL/6 mouse colitis model.)	6.25, 12.5, and 25 μM 5, 10, and 20 mg/kg	Rc	Rc recovers the level of proinflammatory factors (TNF-α, IL-6, IL-1β, ICAM1, and NF-κB)	[139]
	In vivo (Acute-radiation-induced Wistar rat proctitis model.)	20, 40, and 80 mg/kg	Rg3	Rg3-mediated TLR4/MyD88-induced NF-κB inhibition blocks the production of IL-1β and IL-10	[140]
Bone disorder	In vivo (DEX-induced Sprague Dawley rat osteoporosis model.)	1, 6 mg/kg	Rb1	Diminished bone marrow cavity and bone loss in GRb1 treatment GRb1 increases bone formation by increasing AHR and PRELP, which lead to NF-κB deactivation	[147]
	In vivo (Hollow trephine-induced New Zealand rabbit knee OA model.)	30, 100 μg/kg	Rb1	Reduced level of matrix metalloproteinases (MMPs) and TNF-α, TIMP-1, caspase-3, and BAX in Rb1 dose-dependently Rb1 inhibits ROS production as well as cartilage injury, and promotes the formation of proteoglycans and collagen by regulating PI3K/Akt, NF-κB, and MAPK signaling	[148]
	In vitro (*Porphyromonas gingivalis* LPS-stimulated human periodontal ligament cells.) In vivo (*P. gingivalis* LPS-induced Sprague Dawley rat periodontitis model.)	25, 50, and 100 μM 100 μM	Rb3	The expression of IL-6, IL-8, and IL-1β is diminished by Rb3 Rb3 inhibits the inflammatory activation of p38 MAPK, AKT, and NF-κB, especially via downregulating TLR2 expression	[149]

**Table 4 ijms-24-06119-t004:** Therapeutic activities of ginsenosides on cancer models.

Inflammation Site	Test Type	Dose	Type	Molecular Mechanism	Ref.
Breast cancer	In vitro (MDA-MB-231 breast cancer cell line.)	25, 50 μM	Rh1	Rh1 induces the production of mitochondrial ROS, reducing STAT3 activity Rh1-mediated STAT3 deactivation blocks the translocation of NF-κB to the nucleus NF-κB inhibition results in the downregulation of MMP-2 and MMP-9, inhibiting the migration and invasion of MDA-MB-231 cells	[172]
	In vitro (MDA-MB-231 and MCF-7 breast cancer cell lines.)	2, 4, and 6 μM	20(S)-Rh2	20(S)-Rh2 directly binds to Anxa2, inhibiting its function in NF-κB activation Suppressive effects on NF-κB reduced the expression of EMT markers, including MMP-2, MMP-9, Snail, Slug, and TWIST1 20(S)-Rh2 ameliorates the migration and invasion of breast cancer cells	[176]
Lung cancer	In vitro (A549 and PC9 lung adenocarcinoma cell lines.) In vivo (A549 xenograft nude mouse model.)	75, 100, and 125 μM 10, 20 mg/kg	Rk1	Rk1 induces cell cycle arrest in lung cancer cells Rk1 downregulates the NF-κB signaling pathway, leading to a decrease in the PD-L1 expression By lowering PD-L1 expression Rk1 regulates tumor immunity, increasing the apoptosis of lung cancer cells Rk1 lowers the tumor size in a xenograft mouse model with fewer side effects compare to gefitinib	[174]
	In vitro (TGF-β1-treated A549 lung adenocarcinoma cell line.)	Rk1 (45, 50 μM) Rg5 (200, 250 μM)	Rk1, Rg5	TGF-β1 results in the sphere formation and stemness of A549 cells, increasing lung cancer stem cell markers The co-treatment of Rk1 and Rg5 attenuates the translocation of NF-κB, modulating the expression of MMPs and EMT markers Rk1 and Rg5 inhibit the proliferation and EMT of A549 cells	[173]
Osteosarcoma	In vitro (U20S osteosarcoma cell line.)	8, 80 μM	Rh2	Rh2 treatment ameliorates the PI3K/Akt/mTOR signaling axis Through the inhibition of the PI3K/Akt/mTOR signaling axis, Rh2 upregulates IκBα activity, lowering the phosphorylation of p65 Rh2 decreases the migration capacity of U20S cells and induces the apoptosis of U20S cells	[177]

## Data Availability

The data are contained within the article.

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
