# Peer review of "Ginsenosides from Panax ginseng as Key Modulators of NF-κB Signaling Are Powerful Anti-Inflammatory and Anticancer Agents"

_ijms, 2023, doi:10.3390/ijms24076119_

Round 1

Reviewer 1 Report

Please see the attached file with comments for the authors

Reviewer 2 Report

The manuscript from Won Young Jang et al. is a clear and well written review of the antiinflammatory and anticancer effects of ginsenosides. The authors convincigly make the case that many of the therapeutic effects of ginsenoids are mediated by the inhibition of the canonical  NF-kappaB signaling pathway. Bibliography is extensive and accurate. (one typo at the end of line 702). On line 503 and 673:  EGFR stands for epidermal growth factor receptor 

Aa a minor point this reviewer suggests that a better description of the chemical structures of ginsenoids could improve the manuscript: 1) in Figure 1 the meaning or R1 and R2  should be explained; 2) structures of PPD type and PPT type  ginsenoids should be added. 

Reviewer 3 Report

Title: Ginsenosides from Panax ginseng, Key Modulators of NF-kB Signaling as Powerful Anti-Inflammatory Agents

This review paper introduces current studies on the relationship between NF-kB and ginsenosides from Panax ginseng to suggest viable direction for treating inflammatory diseases. The novelty of this review was to search more than 170 valuable articles from different sources like NCBI and Embase to elucidate the therapeutic efficacy of ginsenosides in reducing NF-kB responses and intensify the vital role of ginsenosides in inflammatory diseases and cancers. However, there are several minor concerns as follows:

1.    There are some typos. The manuscript is suggested for English language corrections and improvements.

2.    Please cross-check the references in the list of references and citations in the text.

Reviewer 4 Report

The Cho and coworkers took an effort to present the current knowledge of anti-inflammatory potential of Panax ginseng.  The manuscript contains very interesting and important data and is potentially noteworthy for the readers. It is quite easy to follow, however it requires some modifications before being accepted for publication.

The title should contain also additionally “anti-inflammatory and anticancer agents”  

In the Introduction instead the basic information about NFkB (it is revealed further in the main text) I suggest to present more information about Panax ginseng as a plant and medicine, including its phytoconstituents (structures, quantitative data).

Major comment: there is no quantitative data (concentration/doze/time of incubation/time of treatment) of Panax ginseng describing the studied biological effect. Very often the Authors do not present the type and names of cell lines used in the in vitro studies or tissue origin of cells during in vivo experiments.

I also suggest to present separately data obtained during studies performed with chemically pure compounds (identified in Panax ginseng) or performed with the Panax extract.

What about the impact of P. ginseng on fat tissue, which is also involved in inflammation generation?

I suggest to add information about antioxidant potential of P. ginseng constituents, since oxidative damage strongly activates inflammatory response.

Instead of “microflora” use term “microbiota”; please add more information about the P. ginseng effect on alteration of microbiota and the impact on inflammatory response.

Tables 2/3 – add the names of strains of animals used in studies, as well as the doses/concentrations inducing the biological effect.

Information presented in Chapter 4 can be rather divided in some parts and matched with the observed activity of P. ginseng. The same comment is for Chapter 2.

After each of the chapter I suggest to present figures illustrating the discussed mechanism of P. ginseng biological action. It will strongly enrich presented review.

 Figure 3 resembles rathe graphical abstract – it is too simple for the presented material.

What about the correlation between the P. ginseng concentration/doze causing biological effects observed in vivo and in vitro studies? Please, comment this in the Conclusions section. Give more details about clinical trials, as well as discuss the bioavailability issue.

In summary, the manuscript requires the major revision.

Round 2

Reviewer 1 Report

Please see the attached MS PDF file.

Author Response

Reviewer #1 comments on revised:

Ginsenosides from Panax ginseng, Key Modulators of NF-κB Signaling as Powerful AntiInflammatory Agents

Jang, W.Y., Hwang, J.Y. and Cho, J.Y.

This revision has greatly improved the readability of this paper, and I have no problems now with most of this review. All the necessary corrections I could find were in the first couple of pages, and the only significant problems are with the wording of the gensinoside chemical structure (triterpene) discussion on Page 2. Ocotillol is really a modified PPT derivative, with very little structural similarity to oleanolic acid or its derivatives. Please see the below detailed comments.

Page 1, Abstract, line 16:

“All of the ginsenosides tested targets different steps of the NF-κB signaling pathway ameliorating the symptoms of severe illnesses.” [Sorry, this was partly my fault: the change from “Each ginsenoside…” to “All of the ginsenosides tested” was a singular to plural subject change, so I should have included deletion of the “s” from “targets” in my previous comment].

Page 2, lines 30-31: “…inflammatory diseases diseases such as…”

line 43: “…macrophages, interacts with…”

* Answers: Thanks for your comment and I corrected the points that you mentioned as below.

Line 18: All of the ginsenosides tested target different steps of the NF-kB signaling pathway ameliorating the symptoms of severe illnesses.

   Page 2, lines 30-31: “…inflammatory diseases such as…” (deletion of ‘diseases’)

   Line 43: “…macrophages, interacts with…” (adding ,)

Page 2, lines 79-80: “They are divided into four-ring structure triterpenoid dammarane-types and five-ring structure oleanane-types.” [This statement requires revision, as it is incorrect as a general statement. Dammaranes and oleananes are the only common ring systems that have to date been encountered in (Korean) ginseng, BUT there are many more tetracyclic triterpenoid ring systems such as lanostane, euphane, protostane and tirucallane that like dammarane are found in other plants and fungi. There are even more pentacyclic triterpene ring systems known. Oleanane, ursane and lupane are the 3 most common ones, but there are also many others that are less commonly encountered].

* Answers: Thanks for your comment and I corrected the points that you mentioned as below.

   Most of them have dammarane skeleton, and they are classified into protopanaxadiol (PPD) and protopanaxatriol (PPT) types based on the position and number of hydroxyl groups (see L79-80).

lines 85-87: “Oleanane has two minor structures, the oleanolic acid (OA) and ocotillol (OT). The main difference between them is the number of cyclic structures. The former is a pentacyclic while the latter is a tetracyclic triterpenoid (Figure 1) [44-49]. [These sentences also require revision: only oleanolic acid has an oleanane skeleton. Ocotillol has a tetracyclic dammarane carbon skeleton (with an additional ether ring on the side chain). It probably is derived from a PPT derivative by epoxidation of the double bond at the end of the side chain, then a subsequent concerted opening of the epoxide with the epoxide oxygen displacing the C20 OR3 or OH group to give the 5-membered ether ring, and water providing the oxygen for the teriary alcohol. It should be grouped with the PPT derivatives, NOT Oleanolic acid. The description as tetracyclic or pentacyclic triterpenes only relates to the carbocyclic ring system (which is entirely composed of carbon atoms), not rings that have been formed by oxygen, nitrogen, sulphur or any other heteroatom.]

* Answers: Thanks for your comment and I corrected the points that you mentioned as below. (The (ginsenosides are classified as four types, firstly main types, which has a dammarane structure, divided into PPD and PPT. And ocotillol type is also described as an derivatives of PPT. And the oleanolic acid is explained additionally.)

Lines 84-86: In the case of ocotillol-type, it is derived from PPT, which has tetrahydrofuran ring as a side chain [44]. The other type, the oleanolic acid (OA), has an oleanane skeleton as an aglycone of triterpenoid saponins.

Lines 86-87: this has been deleted.

Reviewer 4 Report

The Authors answered my concerns and improved the manuscript, therefore, in my opinion, the manuscript can be accepted for publication.

Author Response

Thanks very much for your good words.